# β Oscillations of Dorsal STN as a Potential Biomarker in Parkinson’s Disease Motor Subtypes: An Exploratory Study

**DOI:** 10.3390/brainsci13050737

**Published:** 2023-04-28

**Authors:** Yongjie Li, Yuqi Zeng, Mangui Lin, Yingqing Wang, Qinyong Ye, Fangang Meng, Guofa Cai, Guoen Cai

**Affiliations:** 1College of Information Engineering, Guangdong University of Technology, Guangzhou 510006, China; 2Department of Neurology, Fujian Medical University Union Hospital, Fuzhou 350001, China; zengyq@fjmu.edu.cn (Y.Z.);; 3Fujian Key Laboratory of Molecular Neurology, Institute of Clinical Neurology, Institute of Neuroscience, Fujian Medical University, Fuzhou 350001, China; 4Beijing Neurosurgical Institute, Capital Medical University, Beijing 100070, China; fgmeng@ccmu.edu.cn

**Keywords:** deep brain stimulation, dorsal β oscillations, Parkinson’s disease, microelectrode recordings, postural instability gait difficulty, tremor dominant

## Abstract

Parkinson’s disease (PD) can be divided into postural instability and difficult gait (PIGD) and tremor dominance (TD) subtypes. However, potential neural markers located in the dorsal ventral side of the subthalamic nucleus (STN) for delineating the two subtypes of PIGD and TD have not been demonstrated. Therefore, this study aimed to investigate the spectral characteristics of PD on the dorsal ventral side. The differences in the β oscillation spectrum of the spike signal on the dorsal and ventral sides of the STN during deep brain stimulation (DBS) were investigated in 23 patients with PD, and coherence analysis was performed for both subtypes. Finally, each feature was associated with the Unified Parkinson’s Disease Rating Scale (UPDRS). The β power spectral density (PSD) in the dorsal STN was found to be the best predictor of the PD subtype, with 82.6% accuracy. The PSD of dorsal STN β oscillations was greater in the PIGD group than in the TD group (22.17% vs. 18.22%; *p* < 0.001). Compared with the PIGD group, the TD group showed greater consistency in the β and γ bands. In conclusion, dorsal STN β oscillations could be used as a biomarker to classify PIGD and TD subtypes, guide STN-DBS treatment, and relate to some motor symptoms.

## 1. Introduction

Parkinson’s disease (PD) is a progressive neurological disorder of the human nervous system [1]. The incidence rate and progression differences among patients with PD are due to environmental and inherited factors. The exact mechanism of PD is still unclear. PD is widely classified into tremor dominant (TD) and postural instability and gait difficulty (PIGD) subtypes based on clinical research scales [2]. The Unified Parkinson’s Disease Rating Scale (UPDRS) and revised version of the Society for Motor Disorders are usually used for classification [3,4]. Clinical studies have a difference in the rate of progression between TD and PIGD subtypes, with patients with the PIGD subtype progressing more rapidly [5]. Patients with PIGD are at greater risk of motor disorders (e.g., myotonia, gait freezing, and resting tremor) and nonmotor disorders (e.g., dementia, memory loss, and reduced sense of smell) [6].

The advent of microelectrode recording (MER) has led to a better understanding of the subthalamic nucleus (STN) and the identification of possible mechanisms for deep brain stimulation (DBS) [7,8]. MER can be divided into two types of signal: a low-frequency signal known as the local field potential and a high-frequency signal known as a spike. Analysis of these two signals allows the exploration of the electrophysiological mechanisms underlying both TD and PIGD typologies. In previous studies, β oscillations have been strongly associated with the clinical symptoms of PD [9]. During the neuronal activity, β oscillations increase excessively in PD patients with basal ganglia dysfunction, [10,11] and strongly correlated with the severity of motor impairment [12,13]. Levodopa-amine and DBS treatment attenuated β oscillations and improved the clinical symptoms [14,15]. The STN was split into dorsal and ventral sides to examine the effect of β oscillations on PD symptoms in more detail. The dorsal side of STN is the sensorimotor area, which gathers most of the oscillatory activities [16]. Stimulating the dorsal side of STN with DBS can improve the symptoms in patients with PD [16]. In addition, dorsolateral β oscillations can be used to predict the response of patients with PD to DBS and drugs [17]. The spectral analysis of PIGD and TD subtypes on the STN dorsoventral side has not been specifically discussed.

The present study investigated the correlation between the β power spectral density (PSD) of STN in PIGD and TD. Additionally, we further divided the STN into dorsal and ventral parts to study the β PSD of PIGD and TD subtypes. Finally, we analyzed the ability of β PSD in the dorsal STN to distinguish between the two subtypes. These results may provide potential neural markers to differentiate between these two subtypes.

## 2. Materials and Methods

### 2.1. Patient Data

Thirty-five patients with primary PD were prospectively enlisted in the study from February 2019 to July 2022. The inclusion and exclusion criteria for patients with PD are shown in Figure 1. Of the 35 patients, 12 were excluded for the following reasons: 4 patients lost intraoperative transcripts, 2 DBS were globus pallidus internus (GPI) targets (non-STN targets), and 6 had poor microelectrode signal quality. A total of 23 patients (10 males and 13 females) eligible for STN-MER were ultimately included in the study (Figure 1). The Institutional Review Board of the Union Hospital of Fujian Medical University approved the study, and written informed consent was obtained from all participants. According to the TD and PIGD score ratio in UPDRS (ratio ≥ 1.5, TD type; ratio ≤ 1.0, PIGD type), 14 patients were classified into the PIGD type and 9 into the TD type.

### 2.2. Clinical Information

Clinical characteristics, including the age, sex, disease duration, and levodopa equivalent daily dose (LEDD), were recorded. Motor impairment was assessed using Hoehn and Yah (H-Y) staging and the UPDRS Part III in both medication and discontinuation states.

### 2.3. Microelectrode Recording

Of the 23 MERs from the DBS, 17 were recorded by the NeuroNav Physiological Navigation System 4.5.3 (Alpha Omega Engineering Ltd., Nazareth, Israel) and tungsten electrodes (STR-007080-10, Alpha Omega Engineering Ltd.). The other six were recorded using the StealthStation Neuro-Navigation platform (Medtronic Ltd., Minneapolis, MN, USA) and tungsten electrodes (STR-007080-10, Alpha Omega Engineering Ltd.). MERs were filtered with a fourth-order Butterworth band-pass filter (300–6000 Hz). The MER recording started when the electrode arrived 10 mm over the designated target and was advanced in 1 mm increments [18]. Once the electrode tip reached the dorsal border of the STN, the electrode advancement step was decreased to 0.5 mm. The recording ended when the electrode tip reached the substantia nigra. Data greater than 6 s were screened first in the study. Each unit was visually recognized to detect the shape and waveform of the neurons.

### 2.4. Spike Extraction

Spike trains were quantified at an amplitude of ≥4 standard deviations [19,20,21]. The filtered MERs were imported into Neuroexplorer (Micro Bright Field, Inc., Williston, VT, USA) to determine the average firing rate (FR). The asymmetry index (AI) and modified burst index (MBI) of the inter-spike intervals were calculated [21,22]. The signal extraction process is presented in Appendix A.

### 2.5. Spike Power Spectrum Density Estimation

The PSD of the units was used to calculate the oscillation characteristics with Spike 2 (Cambridge Electronic Design, Cambridge, UK). The raw data were rectified, the direct current (DC) was removed with a time constant of 0.5 s down-sampled to 3 kHz, and the noise at 48–52 Hz was removed [23]. The PSD was calculated using the Welch method with a Hanning window. The fast Fourier transform size was 4096, and the frequency resolution was 0.7336 Hz [23]. The PSD of each unit was imported into MATLAB 2018 (The Mathworks, Inc., MA, USA) for normalization, thus obtaining a relative power for each frequency band, that is, θ (3–8 Hz), α (8–13 Hz), β (13–30 Hz), and γ (30–100 Hz). The intraoperative records divided STN into dorsal (0–50%) and ventral (50–100%) sides [17,24].

### 2.6. Background Activity

The first 0.5 ms and last 2.5 ms segments of each spike were replaced with randomly selected spike-free 3 ms segments of the current trace. This new trajectory was defined as a background activity. The new track was observed, and the presence of a spike was checked. The full-wave rectification method was used to remove the DC components [24,25].

### 2.7. Coherence between Spike and Background Activity

The coherence analysis was used to evaluate the extent of frequency correlations between spike trains and background activity. The background activity was introduced using Neuroexplorer, and the coherence analysis was applied. The Welch method was adopted, and the parameters included Hanning Windows and a 50% overlap between windows that produced a 1/2 Hz spectral resolution. The average rate of significant coherence was compared and calculated at each recording depth for the PIGD or TD groups.

### 2.8. Statistical Analysis

SPSS software 26 (IBM Corp., Armonk, NY, USA) and MATLAB 2018 were used for the statistical analysis. The Kolmogorov–Smirnov test was used to determine whether the variables conformed to a normal distribution. For variables that fitted a normal distribution, the Student *t*-test was used. The Wilcoxon rank-sum test was adopted for variables that did not conform to a normal distribution. The significance level was established at 0.05 (two-tailed) and amended based on the number of comparisons [24]. The false discovery rate was introduced in the statistical analysis of each band to prevent type I errors [24]. The Bonferroni multiple-comparison test was used for correction. The Spearman correlation analysis was used to correlate the PSD of the extracted bands and the three eigenvalues with the UPDRS scale. The significance level was *p* < 0.05. Due to the exploratory nature of the study, no correction for multiple comparisons was used.

## 3. Results

### 3.1. Baseline Characteristics

The baseline characteristics of patients with PD are described in Table 1. The patients’ mean age was 61.39 (61.39 ± 6.49) years, and the disease duration was 8.51 (8.51 ± 5.22) years. Men accounted for 43.47% of the sample, and the mean height was 161.60 (161.60 ± 9.74) cm and 61.13 (61.13 ± 11.31) kg, respectively. The TD and PIGD subtypes were comparable according to *p* values (all *p* values were greater than 0.05) (Table 1). The performance of the two subtypes after treatment with DBS is presented in Table 2. Significant variations were observed between the two groups in tremor and axial scores under the medication-off (Med-off) state, and both groups significantly improved after DBS (Table 2).

### 3.2. Comparisons of MBIs, FRs, and AIs

The FR, MBI and AI of two subtypes of neurons are presented in Appendix A. The mean discharge rate in the PIGD and TD groups was 37.62 (37.62 ± 10.11) Hz and 36.00 (36.00 ± 3.91) Hz, respectively, with MBI vales of 1.86 (1.86 ± 0.93) and 1.68 (1.68 ± 0.72), and AI values of 0.27 (0.27 ± 0.06) and 0.26 (0.26 ± 0.03), respectively. No significant differences were found between the three features under different PD subtypes and dorsal and ventral STN locations (*p* > 0.05). (Appendix A)

### 3.3. Spike PSD and Coherence

The PSD comparison of the two PD subtypes is shown in Figure 2. The PSD was divided into four frequency bands: θ, α, β, and γ (Figure 2A,B). The difference between the β power of PIGD (19.80%) and the TD (17.40%) groups was significantly larger (*p* < 0.01). No significant differences were found in the θ, α, and γ bands. The STN was further split into dorsal and ventral areas. A significant difference was found between the PIGD and TD groups in the β band (*p* < 0.001) on the dorsal side. The STN dorsal side of the β power was significantly greater in the PIGD group (22.17%) than in the TD group (18.22%). No significant difference was found between the PIGD and TD groups in the β band (*p* > 0.05) on the ventral side (Figure 2C,D).

The correlation between background activity and spike trains is shown in Figure 3. The comparison is divided into four frequency bands, θ, α, β, and γ, where we focus on the coherence proportion greater than the significance level (95% interval). The comparison is divided into two frequency bands, β and γ. Among these, we focus on a correlation ratio greater than the significance level (95% interval). The TD group had higher coherence in frequency (β: PIGD (17.95%), TD (19.16%); γ: PIGD (69.87%), TD (71.21%)) (Figure 3).

The correlation between the characteristics of the PIGD group under Med-off conditions and the UPDRS scale recorded before DBS surgery is shown in Appendix A. In the Spearman correlation, the mean PSD of the ventral β band was strongly correlated with the H&Y score in the PIGD group (ρ = 0.575, *p* = 0.032). In the Med-off condition, no correlation was found (Appendix A). The correlation between the characteristics of the PIGD group and the UPDRS scale recorded before DBS surgery under Med-on conditions is shown in Table 3. In the Med-on condition, the mean PSD of the θ band correlated with the axial score (ρ = 0.568, *p* = 0.034), and that of the dorsal θ band correlated with the axial score (ρ = 0.552, *p* = 0. 041). The mean PSD of the dorsal α band correlated with the total UPDRS III score (ρ = −0.577, *p* = 0.031), and that of the dorsal β band correlated with the axial score (ρ = −0.582, *p* = 0.029) (Table 3). The correlation between TD group characteristics and the UPDRS scale recorded before DBS surgery under Med-off conditions is shown in Appendix A.

Under the Med-off conditions, the mean PSD on the α, β, and γ bands correlated with the bradykinesia scores in the TD group (α: ρ = −0.695, *p* = 0.038; β: ρ = −0.778, *p* = 0.014; and γ: ρ = −0.778, *p* = 0.014). The mean PSD on the STN dorsal θ band correlated with the bradykinesia score (ρ = −0.695, *p* = 0.038). Moreover, the mean PSD on the STN dorsal β band correlated with both the UPDRS III total score and the bradykinesia score (ρ = −0.678, *p* = 0.045, ρ = −0.770, *p* = 0.015), the mean PSD on the ventral γ band of the STN strongly correlated with the bradykinesia scores (ρ = −0.820, *p* = 0.007), and the MBI characteristics correlated with bradykinesia scores (ρ = −0.720, *p* = 0.029) (Appendix A). The correlation between TD group characteristics and the UPDRS scale recorded before DBS surgery under Med-on conditions is shown in Table 4.

Under the Med-on condition, the mean PSD on the β band correlated with the Bradykinesia score (ρ = −0.762, *p* = 0.017), the mean PSD on the dorsal β band of the STN correlated with the UPDRS III total score, and the bradykinesia score was strongly correlated (ρ = −0.717, *p* = 0.03, ρ = −0.828, *p* = 0.006). The mean PSD on the ventral γ band of the STN correlated with the bradykinesia score (ρ = −0.753, *p* = 0.019), the AI feature correlated with the gait score (ρ = −0.693, *p* = 0.039), and the MBI feature correlated with the bradykinesia score (ρ = 0.728, *p* = 0.026) (Table 4).

The correlation between the UPDRS scale after DBS and MER characteristics of the two subtypes is shown in Appendix A. Appendix A presents the result of the PIGD group, whereas Appendix A presents the result of the TD group. No correlation was found in the PIGD group. The TD experiment revealed a positive correlation between ventral mean PSD and axial score after DBS.

Multiple comparisons of PSD and UPDRS of the dorsal beta in the PIGD group are shown in Appendix A. After multiple comparisons, the axial score (med-off) was still significantly correlated with the characteristic. Multiple comparisons of PSD and UPDRS of dorsal beta in the TD group are shown in Appendix A. After multiple comparisons, the significant correlation between UPDRS part III (med-off) disappeared after multiple comparisons. The significant correlation between Bradykinesia (med-off), Bradykinesia (med-on) and UPDRS part III (med-on) remained.

### 3.4. Distinction between PIGD and TD Subtypes Correlates with β Oscillations on the Dorsal Side of the STN

The STN dorsal β oscillations were used as predictors because they were associated with the differentiation of the two subtypes. Therefore, binary logistic regression was performed with STN dorsal β oscillations as an independent parameter and PIGD and TD as dependent parameters. The equation was as follows:
(1)y=14.415−43.397x
where *y* represents patients with PD with the PIGD subtype, and *x* is the mean PSD of dorsal STN β oscillations. The results showed that the mean PSD of STN dorsal β was a significant predictor (*p* = 0.043), and the ratio (OR) and 95% confidence interval (CI) were 0 (0~0.273). On the contrary, the mean PSD for β oscillations was not a significant predictor (*p* = 0.267). The percentage of correct prediction of the model was 82.6%.

### 3.5. Identifying and Predicting PIGD and TD

The receiver operating characteristic (ROC) curves for both subtypes are illustrated in Figure 4. We used representative ROC curves to investigate the differences in characteristics between the PIGD and TD groups. The mean PSD for STN β oscillations and the mean PSD for dorsal STN β oscillations performed well, with areas under the curve of 0.73 and 0.905, sensitivities of 0.714 and 0.929, and specificities of 0.778 and 0.778, respectively (Figure 4).

## 4. Discussion

This cross-sectional study investigated the PSD and coherence of two PD subtypes, PIGD and TD. First, we confirmed that the PIGD and TD subtypes differed in their β oscillations. Second, further subdivision of the STN into dorsal and ventral regions to explore the effect of dorsal and ventral β oscillations demonstrated a strong correlation between the dorsal β oscillations of the STN and the PIGD and TD subtypes. Finally, ROC curves and binary logistic regression demonstrated that the STN dorsal β oscillations better distinguished the two PD subtypes. Furthermore, the consistency analysis between the two subtypes revealed more consistency in the TD group β and γ. These results suggest that the β oscillations on the dorsal side of the STN had potential predictive values in delineating the two subtypes.

### 4.1. Dorsal β Oscillations in Relation to Two Subtypes of PIGD and TD

A previous study found differences between the two subtypes on the low-β band. The β oscillatory active neurons correlated with the electromyography of the limbs in rigid and bradykinetic PD [27]. Similar results were found in this study, where the PIGD and TD differed in STN β, and the PIGD group showed greater oscillations than the TD group. In previous studies, β oscillations are a useful and reliable feature of the dorsal STN of patients with PD. Stimulating β oscillatory active neurons and θ oscillatory neurons in the STN dorsal region is an effective way to alleviate PD symptoms [27,28]. This study confirmed that dorsal STN β oscillations are a reliable feature; they have a reliable role in discriminating PIGD and TD subtypes. Previous studies showed strong β oscillations in the dorsal region of the STN and strong γ oscillations in the ventral region of the STN. Similar results were found in this study, particularly in the PIGD group. Previous studies found that the β oscillations were greater in the STN dorsal region than in the ventral region [29]. The present study exhibited consistent results, with the dorsal side being larger than the ventral side in both the TD and PIGD groups.

### 4.2. Coherence Analysis of the Two PIGD and TD Subtypes

This analysis was not performed in either subtype. One study found a low rate of interneuronal coherence in the β band, predominantly present in patients with intense tremors [30]. Another study combining pallid layer local field potential recordings with scalp electroencephalogram (EEG) in patients with PD [31] showed a high β coherence between the pallid layer and the cortex in patients with motor rigidity PD, which decreased as levodopa improved symptoms [32]. All these studies demonstrated that the coherence was stronger in the β band in patients with TD-type PD, which was consistent with the findings of the present study.

### 4.3. Correlation of PSD and UPDRS Scores

We studied each band’s extracted PSDs and made correlations in the two fractal types. In the TD group, under both conditions, the β oscillations significantly correlated with the bradykinesia scores but not with tremor and gait, consistent with previous findings showing a significant correlation between the β band power and symptoms of stiffness and weakness estimated in the Med-off state [33,34]. In the PIGD group, the β oscillations of the dorsal STN in the Med-on condition correlated with the axial scores. This was also consistent with a recent study on the association of various regions in STN with PD symptoms [33]. Previous studies examining patients who responded well to anti-Parkinsonian drugs showed that axial features and motor retardation were partially responsive to levodopa, with effects generally close to those obtained with STN stimulation [26,35]. Based on this study, we speculated that the difference in response to the drug led to some differences in the results in the patients with PIGD and TD in the current trial.

### 4.4. Limitations

This study had some limitations. The patients included in this study were operated on with different instruments during the procedure. During the procedure, bleeding due to anesthesia or electrode insertion might have affected the results. Therefore, the surgical procedure was strictly implemented to minimize other factors in the experimental data. No spike sorting was performed when analyzing AI, MBI, FR, PSD, and the correlation between the two subtypes, so our results are insufficient to conclude at the level of individual units. In our follow-up data, some patients were followed for 1 month and some for 6 months. Therefore, we could not perform systematic correlation analyses for different follow-up times. Due to the limited sample size (N = 23) in this study, validating our conclusion using a larger sample size is necessary. The length of MER recorded during surgery is usually smaller than that recorded after implantable IPG surgery due to surgical limitations, which may have affected our results. Our patients are mainly diagnosed with mid- to late-stage PD (H-Y stage > 3 stage), accompanied by severe symptoms of Parkinson’s disease, resulting in a relatively high severity of TD subtypes in our study.

## 5. Conclusions

In this study, we demonstrated that the PIGD and TD motor phenotypes of PD exhibited different dorsal side β dynamics. We also showed that the dorsal β oscillation was negatively correlated with axial scores in the PIGD group and with motor retardation scores in the TD group. These findings contribute to a better understanding of the pathophysiological aspects of PD and might establish new electrophysiological markers.

## Figures and Tables

**Figure 1 brainsci-13-00737-f001:**
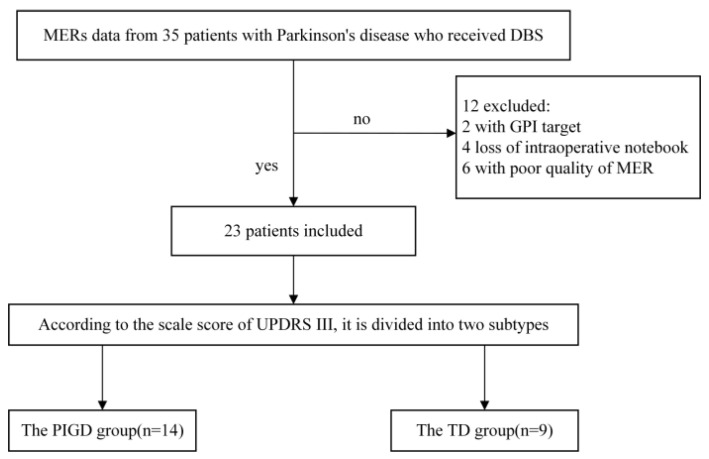
Inclusion and exclusion of patients with PD. MERs, microelectrode recordings; DBS, deep brain stimulation; GPI, globus pallidus internus; PIGD, postural instability and gait difficulty; TD, tremor dominant.

**Figure 2 brainsci-13-00737-f002:**
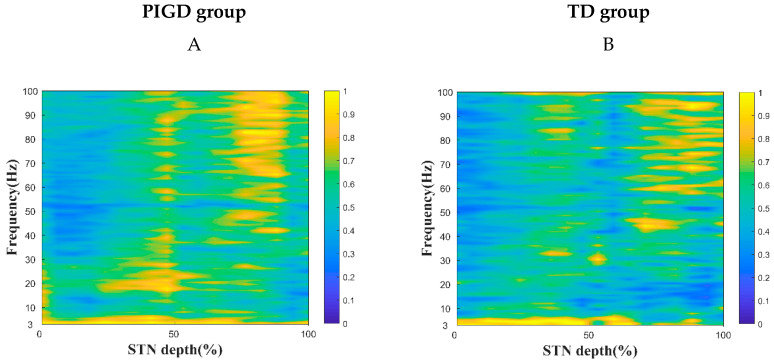
Comparative analysis of two subtypes of power spectral density (PSD) in the subthalamic nucleus (STN). Variation in the PSD plot shows that the PIGD group oscillated at higher energy values in the β band. (**A**) The PIGD subtype microelectrode records data from the dorsal and ventral STN. (**B**) The TD subtype microelectrode records data from the dorsal and ventral STN. A two-tailed Wilcoxon rank-sum test was used to compare each band between the PIGD and TD groups (with 250 MERs in the PIGD group and 129 MERs in the TD group), and divided into the upper and lower sides of the STN nuclei. Significant differences were found between the PIGD and TD groups in the dorsal part of the STN. (**C**) Comparison of Spectral Sizes between Two Subtypes in Four Frequency Bands. (**D**) Comparison of spectral sizes of two subtypes of STN on the dorsal and ventral sides in four frequency bands. ** Indicates *p* < 0.001 in the PIGD and TD groups.

**Figure 3 brainsci-13-00737-f003:**
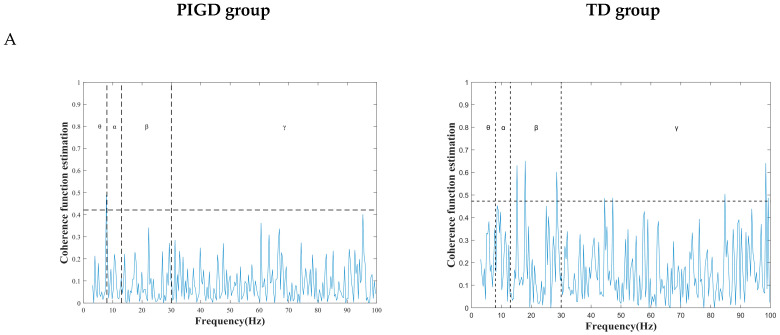
Comparison of correlations between the PIGD and TD groups for each power band. (**A**) Coherence is quantified as a significant correlation between spike and background activity, with dashed lines indicating significant coherence at 95% confidence intervals. (**B**) Comparison chart of significant coherence percentage of two subtypes in β and γ.

**Figure 4 brainsci-13-00737-f004:**
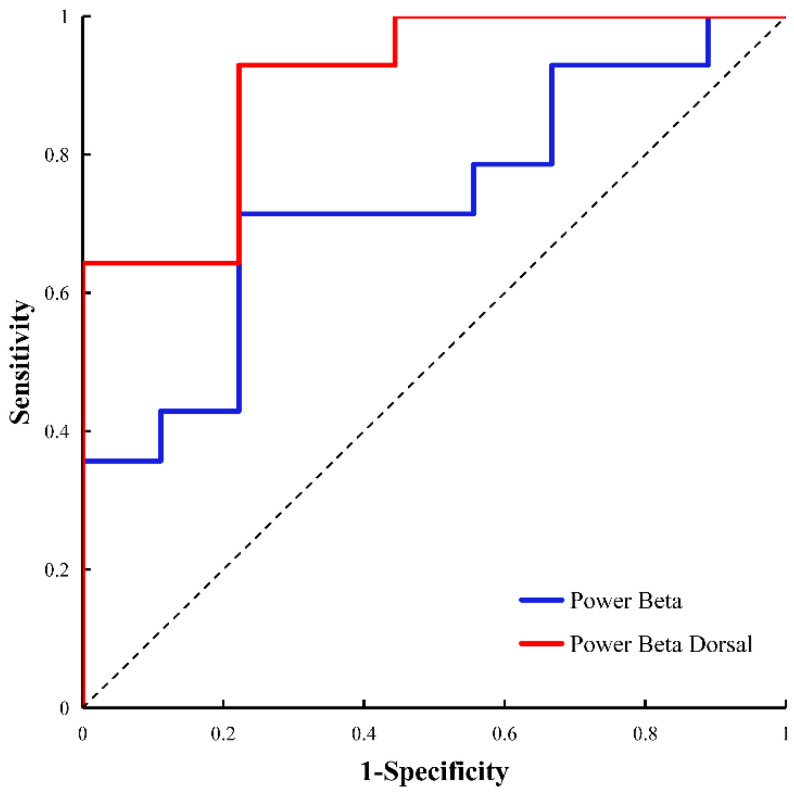
ROC curves for PIGD and TD differentiated using the power spectral density (PSD) in the β band: PSD in the β band (blue) and PSD in the dorsal beta band (red).

**Table 1 brainsci-13-00737-t001:** Baseline characteristics.

	PIGD Group (*n* = 14)	TD Group (*n* = 9)	*p* Value
Age (years), ${\overset{-}{X}}^{}\pm SD$	61.29 ± 4.93	61.56 ± 8.73	0.92 ^a^
Age at onset of PD (years), ${\overset{-}{X}}^{}\pm SD$	52.36 ± 5.08	53.65 ± 10.49	0.73 ^a^
Height (cm), ${\overset{-}{X}}^{}\pm SD$	160.14 ± 9.02	163.89 ± 10.92	0.56 ^a^
Sex (male/female)	5/9	5/4	0.34 ^b^
Body mass index (kg/m^−2^)${\overset{-}{X}}^{}\pm SD$	23.014 ± 4.76	23.978 ± 3.06	0.59 ^a^
Duration of PD (years), ${\overset{-}{X}}^{}\pm SD$	8.91 ± 6.03	7.89 ± 3.89	0.65 ^a^
Weight (kg), ${\overset{-}{X}}^{}\pm SD$	59.29 ± 13.47	64.00 ± 6.46	0.34 ^a^
LEDD (mg), ${\overset{-}{X}}^{}\pm SD$	1003.43 ± 359.16	1002.78 ± 263.93	0.99 ^a^
UPDRS III off, ${\overset{-}{X}}^{}\pm SD$	53.93 ± 10.916	49.00 ± 13.295	0.90 ^a^
H&Y stage, ${\overset{-}{X}}^{}\pm SD$	3.32 ± 0.66	2.88 ± 0.54	0.12 ^a^

Numbers indicate means ± standard deviations (SD) or number (percentage). The LEDD was calculated according to drug correspondences proposed by Tomlinson et al. [26]. Differences between the two groups were analyzed using the *t*-test ^a^ or Pearson’s chi-squared test ^b^. PD, Parkinson’s disease; LEDD, Levodopa equivalent daily dose; UPDRS III, Part 3 Unified Parkinson’s Disease Rating Scale significance was recognized when *p* < 0.05.

**Table 2 brainsci-13-00737-t002:** Preoperative and postoperative medication and DBS effectiveness between PIGD and TD groups.

		Part III	Tremor	Rigidity	Bradykinesia	Axial
Med off	PIGD	53.93 ± 10.91	3.36 ± 3.22	12.64 ± 4.60	20.50 ± 3.61	13.36 ± 3.71
	TD	49.00 ± 13.29	7.56 ± 4.66	10.89 ± 4.51	18.33 ± 3.50	9.00 ± 3.70
Med on	PIGD	24.57 ± 7.40	1.14 ± 1.91	4.71 ± 3.60	10.36 ± 3.97	5.86 ± 1.79
	TD	24.11 ± 9.88	0.56 ± 0.72	6.89 ± 3.95	9.89 ± 6.15	4.56 ± 2.18
DBS on	PIGD	21.07 ± 10.79	0.00 ± 0.00	3.86 ± 2.17	9.14 ± 4.40	5.07 ± 3.09
	TD	18.56 ± 7.24	1.00 ± 1.58	2.67 ± 3.46	8.11 ± 2.36	3.22 ± 1.92
*p* value	Med off	0.34	0.01 *	0.37	0.17	0.01 *
	Med on	0.90	0.39	0.18	0.82	0.13
	DBS on	0.54	0.09	0.32	0.52	0.14

* represents a statistical correlation (*p* < 0.05).CI, confidence interval; DBS on, deep brain stimulation on; H&Y, Hoehn and Yah; Med off, medication off; Med on, medication on; PIGD, postural instability and gait difficulty; TD, tremor dominant; UPDRS, Unified Parkinson’s Disease Rating Scale. Data are expressed as mean (95% CI). The Student *t*-test was used for statistics for PIGD and TD under Med off, Med on, and DBS on.

**Table 3 brainsci-13-00737-t003:** Results of the correlation analysis between MER characteristics and clinical characteristics in the PIGD group (Med-on).

MER Features	UPDRS III (Med-on)					
	UPDRS III	Tremor	Rigidity	Bradykinesia	Axial Symptoms	Gait
Power_Theta	−0.02/0.946	−0.264/0.362	−0.383/0.176	0.401/0.155	0.568/0.034 *	0.203/0.487
Power_Alpha	−0.123/0.674	−0.081/0.782	−0.229/0.43	−0.029/0.922	−0.272/0.348	−0.051/0.864
Power_Beta	0.002/0.994	−0.025/0.933	0.025/0.934	−0.158/0.589	−0.353/0.216	−0.203/0.487
Power_Gamma	0.194/0.507	−0.133/0.65	0.069/0.815	0.301/0.296	0.466/0.093	0.051/0.864
Power_Dorsal_Theta	−0.172/0.557	−0.225/0.44	−0.225/0.439	0.118/0.688	0.552/0.041 *	0.456/0.101
Power_Dorsal_Alpha	−0.577/0.031 *	−0.447/0.109	−0.523/0.055	−0.301/0.296	0.025/0.933	0.152/0.604
Power_Dorsal_Beta	−0.081/0.782	0.153/0.602	−0.022/0.94	−0.247/0.394	−0.582/0.029 *	−0.506/0.065
Power_Dorsal_Gamma	0.194/0.507	−0.133/0.65	0.105/0.722	0.234/0.421	0.502/0.067	0.051/0.864
Power_Ventral_Theta	0.115/0.697	0.02/0.947	−0.352/0.217	0.486/0.078	0.254/0.382	−0.101/0.73
Power_Ventral_Alpha	0.152/0.604	0.192/0.51	−0.111/0.705	0.216/0.458	−0.201/0.49	−0.203/0.487
Power_Ventral_Beta	0.059/0.84	−0.22/0.451	0.147/0.616	−0.145/0.621	−0.256/0.377	−0.051/0.864
Power_Ventral_Gamma	0.119/0.685	−0.027/0.927	−0.116/0.693	0.303/0.292	0.192/0.51	−0.017/0.837
Firing rate	−0.163/0.578	0.076/0.795	−0.082/0.779	−0.12/0.682	0.238/0.413	0.238/0.358
Asymmetry_Index	−0.256/0.378	−0.168/0.566	−0.087/0.768	−0.301/0.296	0.274/0.343	−0.341/0.263
MBI	−0.269/0.353	0.36/0.206	−0.232/0.426	−0.105/0.722	0.1/0.735	−0.051/0.864

Values are presented as rho (*ρ*)/*p* value; * represents a statistical correlation (*p* < 0.05). UPDRS, Unified Parkinson’s Disease Rating Scale.

**Table 4 brainsci-13-00737-t004:** Results of the correlation analysis between MER characteristics and clinical characteristics in the TD group (Med-on).

MER Features	UPDRS III (Med-on)					
	UPDRS III	Tremor	Rigidity	Bradykinesia	Axial Symptoms	Gait
Power_Theta	−0.083/0.831	0.484/0.186	−0.05/0.898	−0.276/0.472	0.604/0.085	−0.26/0.5
Power_Alpha	−0.133/0.732	0.335/0.378	−0.05/0.898	−0.351/0.354	0.604/0.085	−0.173/0.656
Power_Beta	−0.617/0.077	−0.149/0.702	−0.603/0.086	−0.762/0.017 *	0.477/0.194	−0.007/0.974
Power_Gamma	−0.567/0.112	−0.261/0.498	−0.527/0.145	−0.644/0.061	0.621/0.074	0.346/0.361
Power_Dorsal_Theta	−0.133/0.732	0.335/0.378	−0.05/0.898	−0.351/0.354	0.604/0.085	−0.173/0.656
Power_Dorsal_Alpha	−0.1/0.798	0.41/0.273	−0.033/0.932	−0.301/0.431	0.57/0.109	−0.26/0.5
Power_Dorsal_Beta	−0.717/0.03 *	−0.261/0.498	−0.603/0.086	−0.828/0.006 **	0.451/0.223	0.26/0.5
Power_Dorsal_Gamma	−0.467/0.205	−0.186/0.631	−0.628/0.07	−0.502/0.168	0.383/0.309	0.173/0.656
Power_Ventral_Theta	0.033/0.932	0.596/0.09	−0.142/0.715	−0.008/0.983	0.57/0.109	−0.346/0.361
Power_Ventral_Alpha	0.2/0.606	0.634/0.067	0.176/0.651	0.033/0.932	0.613/0.079	−0.173/0.656
Power_Ventral_Beta	−0.183/0.637	0.298/0.436	−0.109/0.781	−0.544/0.13	0.145/0.71	−0.346/0.361
Power_Ventral_Gamma	−0.533/0.139	−0.261/0.498	−0.377/0.318	**−0.753/0.019 ***	0.417/0.264	0.173/0.656
Firing rate	−0.517/0.154	0.41/0.273	−0.444/0.232	−0.536/0.137	−0.383/0.309	−0.433/0.244
Asymmetry_Index	−0.217/0.576	0.037/0.924	−0.251/0.515	−0.368/0.33	−0.358/0.345	−0.693/0.039 *
MBI	0.55/0.125	0.484/0.186	0.285/0.458	0.728/0.026 *	0.06/0.879	−0.26/0.5

Values are presented as rho (*ρ*)/*p* value; * represents a statistical correlation (*p* < 0.05); ** represents a statistical correlation (*p* < 0.01). UPDRS, Unified Parkinson’s Disease Rating Scale.

## Data Availability

The data that support the findings of this study are available from the corresponding authors upon reasonable request.

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
