# Peer review of "β Oscillations of Dorsal STN as a Potential Biomarker in Parkinson’s Disease Motor Subtypes: An Exploratory Study"

_brainsci, 2023, doi:10.3390/brainsci13050737_

Round 1

Reviewer 1 Report

This study investigated potential neural markers in the Subthalamic nucleus (STN) to differentiate between two subtypes of Parkinson's disease (PD). By analyzing beta oscillations in the dorsal and ventral side of the STN during deep brain stimulation (DBS), the study found that the β power spectral density in the dorsal STN was the best predictor of PD subtype, with an accuracy of 82.6%. Additionally, the study found that the PIGD group had greater coherence in the θ and α bands compared to the TD group. Authors concludes that dorsal STN β oscillations can be a useful biomarker to classify PD subtypes, guide STN-DBS treatment, and relate to motor symptoms. These findings provide insight into the pathophysiological aspects of PD and suggest potential electrophysiological markers for the different subtypes. The work is well done, organized and the conclusion supported by the signal analysis.

Author Response

Thanks for the positive comments on our manuscript.

Reviewer 2 Report

β oscillations of dorsal STN as a potential biomarker in Parkinson’s disease motor subtypes

The present study investigated the correlation between STN βpower spectral density in PIGD and TD type in PD. The authors also investigated the correlation between dorsal and ventral STN βpower spectral density and clinical features. The author also examined the ability of β power in the dorsal STN to distinguish PIGD and TD.

 The authors concluded that 1) PIGD and TD subtypes differ in their β oscillations, 2) strong correlations exist between the dorsal STN β power and PIGD/TD subtypes, 3) the dorsal STN β power have potential predictive value in delineating the two subtypes.

The followings are my comments to the authors.

1)    Although the authors classified the TD and PIGD subtype according to the score ratio in UPDRS, no significant differences were found in disease duration, LEDD, and the score of UPDRS Part 3 between TD and PIGD types. However, most neurologists might recognize TD subtypes as having longer disease duration, and milder motor symptoms with fewer LEDD than PIGD. Therefore, the clinical features of TD subtypes represented in this study might be different from the generally accepted clinical features of TD subtypes.

2)    The authors examined the ability of β power in the dorsal STN to distinguish TD and PIGD subtypes and concluded that the dorsal STN β power has potential predictive value in delineating the two subtypes. However, because the classification between PIGD and TD subtypes is based on clinical features, I cannot understand the clinical efficacy of using the dorsal STN β power in distinguishing PIGD and TD subtypes. 

Reviewer 3 Report

In this study, the authors retrospectively analyzed the MER during the insertion of electrodes in STN for DBS in 23 PD patints. The patients were, based on UPDRS motor scores then divided into two groups PIGD (14 patents) and TD (9 patients). The main findings of the study were: 1. The PSD of dorsal STN 24 β oscillations was greater in the PIGD group than in the TD group (22.17% vs. 18.22%; P < 0.001). 2. The PIGD group was in greater coherence in the θ and α bands compared to the TD group. The authors conclude that the dorsal STN β oscillations can be used as a biomarker to classify PIGD and TD subtypes, guide STN-DBS treatment, and relate to some motor symptoms.

1. I have not found the information regarding whether the authors used the UPDRS scores before the operation OFF medication? Related to this, it would be nice to see if there are any corrlations of the MER and the motor scores after the operation.

2. One thing that is missing and needs to be discussed is the follwoing: the MER during DBS opeartion are very short. As such, MERs are very variable and might not always reflect the exact motor condition of the patinent. Namely, sometimes during recordings we see very akinetic patients in whom we do not see a good STN signal, and vice versa – a very good STN signal in patients who are not doing that bad. Furthremore, the authors compared these short MERs with the motor condition of tha pateints which is a refclection of the chronic, progressive disease process. This is a limitation of the study and needs to be further discussed and aslo mentioned in the Limitation section. Nowadays it is possible to analyse chronically recorded DBS from certain DBS systems. It would be a good idea to record the LFP-s after the operation and correlated these t the current motor condition of the patinets and to the motor condition before the operation and MER during the operation. 

3. The number of patients is quite low, the number of comparisons very high. You than subivided the patients into two additional groups (PIGD and TD). Did the authros used any correction method fo multiplce statsictical comparisons? The low number of patients is also a limitation of the study and this needs to be stressed out in Limitation section.

4. It might be a good idea to chage the name of the paper from »β oscillations of dorsal STN as a potential biomarker in Parkinson’s disease motor subtypes« into »β oscillations of dorsal STN as a potential biomarker in Pakinson’s disease motor subtypes: An exploratory study« as this study is very exploratory in nature. 

5. I do not really see what is the point in looking at the coherence between frequency bands, hence also the finding of the PIGD group being in greater coherence in the θ and α bands compared to the TD group seems to be vague. You say – theta correlates to faster response times, which you did not tested in your study, and again, intuitivelly one would expect that PIGD patinets have slower response times than TD patients. You conclude that »In our study, patients with PIGD type had 308 more severe dystonia, and it was also evident from the results of the coherence analysis 309 that PIGD was stronger than TD in both the θ and α frequency bands«, but you do not provide any meassure of dystonia. In addition, we are really far from declairing theta band a marker for dystonia. I would suggest that you skip the part on coherence in theta and alpha as this does not adds anything on the main finding related to the beta band.

Round 2

Reviewer 2 Report

Although the authors responded to my question that the "UPDRS score of the PIGD subtype is higher than that of the TD subtype, it is consistent with your statement that the TD subtype has milder motor symptoms and LEDD is smaller than the PIGD subtype", the difference in the score of UPDRS part 3 was subtle and not statistically significant. I think that both the PIGD group and TD group in this study showed rapid progression because the disease duration was 8.91 years for the PIGD group and 7.89 years for the TD group, respectively. However, the typical TD subtypes usually show milder motor symptoms compared to the PIGD type. Although some papers reported that the duration of TD subtypes was shorter than the PIGD type, I think that the severity of TD subtypes (UPDRS Part3 =49.0 during off-medication, disease duration=7.89 years) was relatively high in this study, which might be the limitation of this study.

Author Response

Thanks for the constructive comments.

Line 334, “Our patient is mainly diagnosed with mid to late stage PD (H-Y stage>3 stage), accompanied by severe symptoms of Parkinson's disease, resulting in a relatively high severity of TD subtypes in our study.” was added

Reviewer 3 Report

The authors have now imprved the paper.

Just one more thing regarding the multiple comparisons. You obviously did not use any correction method for multiple comparisons, which is perfectly all right as you conducted an exploratory study. However, this needs to be excplicitely stated in the Methods section as: "Due to the exploratory nature of the study no correction for multiple comparisons was used".

Author Response

Thanks for the constructive comments.

Line 141, “The FDR multiple test method was used to correct the P value.” were corrected as “Due to the exploratory nature of the study no correction for multiple comparisons was used.”